# Analysis of the Lifetime of Neural Implants Using In Vitro Test Structures

**DOI:** 10.3390/s23146263

**Published:** 2023-07-10

**Authors:** Jürgen Guljakow, Walter Lang

**Affiliations:** Institute for Microsensors, Actuators and Systems (IMSAS), University of Bremen, Otto-Hahn-Allee 1, 28359 Bremen, Germany; wlang@imsas.uni-bremen.de

**Keywords:** accelerated lifetime test, neural implant, polyimide, Weibull

## Abstract

The aim of this work was to measure the lifetime of neural implant test samples at two different temperatures, using a method that allows the precise measurement of the sample lifetime, further analysis with the use of Weibull statistics, and examination of the applicability of the Van’t Hoff rule. The correct estimation of the lifetime of neural implants is important to avoid preliminary failures, when used in humans. The novelty lies in the precise data due to the measurement approach, the application of the Weibull statistics to neural test samples, and the examination of the Van’t Hoff rule’s applicability to the longevity of polyimide-based neural implant samples. Several samples that consisted of interdigitated gold strands, encapsulated in polyimide were soaked in ringer solution. One batch was soaked at a temperature of 37 °C, and another was soaked at a temperature of 57 °C. Voltage was applied and measured to identify the occurrence of failures. The long-term experiment was stopped after 458 days for the samples at 37 °C and 423 days for the samples at 57 °C, with several samples still being intact at both temperature levels. The time to failure was measured and used to identify the Weibull parameters that would describe the behavior of the samples. The median lifetime of the samples changed from 363 days at 37 °C to 138 days at 57 °C. The scale and shape factor changed from 396 and 3.7 at 37 °C to 138 and 2 at 57 °C, respectively. The measured mean, median times, and Weibull scale factors were lower than expected from the Van’t Hoff rule. The use of the Van’t hoff rule with $2^{\Delta T/10{}^{\circ}C}$ for accelerated lifetime tests would lead to an estimation of longer lifetimes than realistic. A reaction rate constant around 1.47 appears more appropriate. While a fourfold difference in lifetime would be expected, only a 2.65-fold difference in the median lifetime and a roughly 2.2-fold difference in the mean and Weibull scale factor were observed. The shift of the Weibull shape parameter from 3.7 at 37 °C to 2 at 57 °C with rising temperatures was observed, indicating differences in failure reasons and stronger aging at lower temperatures. The used method is simple to apply and interpret and allows for a precise anticipation of sample lifetimes.

## 1. Introduction

### 1.1. Accelerated Lifetime Tests

Accelerated lifetime tests are used when it is necessary to find out which lifetimes can be anticipated [1]. During those tests, the samples are exposed to conditions that are harsher than the conditions thee samples are usually subjected to. As higher temperatures contribute to a shorter lifespan, the test samples for neural implants will be immersed into a liquid at higher temperatures than the body’s temperature, in order to simulate longer timespans. To calculate the possible lifetime, the Van’t Hoff rule is used. That rule states that for every 10 degree rise in temperature, the simulated lifetime doubles [1,2]. Despite its ubiquitous use, there are uncertainties tied to this approach, considering that the 10 °C rule applies when the initial temperature is 25 °C at a certain activation energy [2]. The need to assess the applicability of the Arrhenius law was described in [2]. A comparison between in vitro and in vivo long-term experiments was described in [3]. The hitherto used test procedure is the EIS, electrical impedance spectroscopy, measuring the impedance and phase of the sample over time [4,5,6]. The changes in those parameters over time gave hints on the deterioration of the sample quality with the increase in the time spent immersed in the solution [4]. The samples underwent measurements in predetermined timeframes. Instead of measuring the impedance after fixed timeframes, in this work, the voltage over a 1kOhm resistance was surveyed over time, by an Arduino, Italy, [7], which allows a resolution of one measurement a second or higher. The design selected for the experiments in this work was interdigitated structures in polyimide (PI). A similar design was used by [8,9]. As soon as a delamination and an intrusion of the surrounding solution takes place, the rise in voltage indicates a current, the loss of insulation, and thus a failure.

### 1.2. Neural Implants

Several accelerated lifetime tests were undertaken with parylene-C (PPX-C) samples, that were meant to simulate the behaviour of neural implants [4,5]. A common measuring method is electrical impedance spectroscopy (EIS) [3,10] or [11], which offers a broad description. With this method, measurements take place after predefined time periods. Another approach is the measurement of resistance over time, where the samples are immersed in a hot bath [12]. In several cases, Utah Arrays were used for the experiments [13,14]. Likewise, there were more sophisticated experiments, where higher temperatures, PBS, and H${}_{2}$O${}_{2}$ were used [15,16]. Another polymer, having a similar application is polyimide. Here, tests were conducted to evaluate the changes in the mechanical properties after the polyimide was subjected to accelerated lifetime tests [17]. Hereby, several different polymers were tested, among them U-Varnish-S, the same as that used in this work. The experiments took place in PBS solution. For evaluation of the accelerated lifetime tests, the Van’t Hoff rule or Arrhenius law is often used or modified [1,2] ([18], p. 87, [19], p. 89). An application of the 10-degree rule to neural implants, although for wire implants in this case, is found in [20]. In particular, [1,2] mention that validation of the calculated data is necessary. Thus, this work evaluates the assumptions laid down by the Van’t Hoff rule, that the reaction rate doubles every time the temperature increases by 10 °C. This rule is expressed by the equation
(1)$$A\left(t\right)=2^{\frac{\Delta T}{10{}^{\circ}C}},$$
where ΔT is the difference in the temperature between the reference temperature and the temperature at which the accelerated aging test is conducted. This equation, used in the norm ASTM F1980, describes accelerated aging and is described as a conservative approach in [21]. Factors other than the reaction rate constant of two can be used, although a higher factor could lead to predictions of longer lifetimes than realistically achievable [21]. Unlike with the EIS, where measurements are undertaken at predetermined times, the measurement takes place continuously, with about one measurement a second. This approach permits pinpointing the exact time a failure occurs. The samples in this work were fabricated as foils, where PI was deposited on silicon, layered with gold, structured, and covered with another PI layer, to be structured again and be prepared for the measurement. The production process is described in detail in [7]. As the basis for the analysis, the Weibull distribution was used.

### 1.3. This Work

This paper is the followup paper to [7], where the failure reasons for the samples that replicated the behavior of possible neural implants were analyzed. The sample preparation and test method are described in detail in [7]. The novelty of this work lays in the application of a measuring method that allows obtaining precise data on sample lifetimes and using that data for further statistical analysis, while at the same time examining the applicability of the Van’t Hoff rule. Earlier works made use of the EIS method for the estimation [3,22] of the sample quality. Hereby, the change in impedance was interpreted as a deterioration of the samples. The measurements of the impedance were conducted regularly. An apparatus for the EIS measurement of statistically relevant numbers of neural implants was presented in [11]. For this work, another approach was chosen that was also tried inhouse [8]. Hereby, the voltage was measured. An increase in voltage above a certain level is interpreted as the sample’s failure. The structure is described in detail in [7]. That approach allows continuous measurement to identify a precise sample lifetime. These measurements are used for further statistical analysis, where the standard deviation and the Weibull distribution are used. A second aim was to analyze how realistic the assumptions about the achievable lifetimes are, which are based on the Van’t Hoff rule. It was expected that the lifetimes would be four times longer for the samples tested at 37 °C, compared to the samples tested at 57 °C. A similar distribution of lifetimes would be expected for both temperature levels. While a change in the scale factor was expected due to the increased temperature, no change in the shape factor was expected. For the accelerated lifetime tests, various conditions were used with temperatures of up to 90° in saline [9], H${}_{2}$O${}_{2}$[16], or application of ultrasonic treatment [8]. Less harsh conditions, like the use of a saline solutions at 60 °C [22], can also be found. As the work of Rubehn has shown, soaking of the polyimide UPILEX-S in saline at a temperature of 60 °C, which was also used in this work, does not lead to changes in the mechanical properties. Trials at 85 °C in a saline solution led to changes in the mechanical properties. Due to that reason and to experiences inhouse [22], a reference temperature of 57 °C was chosen for the accelerated test. The difference of 20 °C allows for easy comparison between the samples held at body temperature and the samples at the elevated temperature. With no further stressors, such as H${}_{2}$O${}_{2}$, etc., only the influence of the temperature was considered. That approach simplifies the comparison between the results and the predictions based on the Van’t Hoff rule. The results may contribute to safer implants, as it allows providing precise information on the sample lifetimes and more precise predictions of the sample lifetimes. The overall problem of the lack of insight about the deterioration of polymer-based neural implants is addressed in [23]. That work provides a wide overview of the used materials and failure reasons of thin-film-based neural implants. While [7] aimed to provide insight into the failure reasons, this work aims to provide a statistical analysis of the sample lifetimes.

## 2. Materials and Methods

### 2.1. Sample Preparation

As precursor, the polyimide varnish U-Varnish-S by UBE, Japan, was used. The polyimide varnish was cured in a vacuum in a curing process that took six hours, within which the temperature rose step-wise from room temperature to 450 °C and was lowered again down to room temperature. In the next step, gold was deposited and structured, and a second layer of PI was applied, which was cured in the same way as the first layer of the PI. Sample preparation and the electric schematics of the build-up were described in detail in [7]. The samples were contacted with conductive glue and put into vials, which were later sealed with epoxy and filled with ringer solution. In total, 16 samples were put into solution and held at a temperature of about 37 °C. A second set of 23 samples was put into a 57 °C bath. The vials that contained the samples were placed in crystallizing dishes that were filled with polyethylene glycol (PEG). The temperature of the PEG was measured with a thermometer, and the heaters were set up accordingly.

A voltage of 5 V was applied to the samples. Each sample was connected in series with a 1 kOhm resistance. The voltage difference was measured over the 1 kOhm resistance. If the sample was still intact, there was no voltage drop. If the sample broke, a sharp increase in the voltage to a level between 4 V and 5 V was observed. The time between the start of the experiment and the breakage of the sample was recorded. The temperature varied between 54 °C and 60 °C; thus, it is referred to as 57 °C in the rest of the text. Arduinos were used for the voltage measurement.

The state of the samples before and after the trial can be seen in Figure 1. Figure 1a shows the state of the sample before the test, without any damage. Figure 1b shows the state of the sample after failure. The delamination and intrusion of the surrounding liquid into the sample on one side is clearly visible. Such a state is common after failure. A detailed description of the failures can be found in [7].

### 2.2. Statistical Analysis of Measurement

For the analysis of the lifetimes, the Weibull function was used, which is a function that describes the reliability of samples with a weakest-link-in-the-chain approach, where a failure occurs in the whole when a failure occurs in any element of the whole, going back to the work of Waloddi Weibull [24]. It is a function that is used in reliability or electrical engineering to describe the reliability or material parameters in the case of insulators. The cumulative distribution function has the form: (2)$$F\left(x\right)=1-e^{-{\left(\frac{x}{\lambda }\right)}^{k}},$$
where λ is the scale parameter, and k is the shape parameter. The Weibull distribution finds use in electric engineering ([25], p. 55), i.e., in the analysis of an insulator breakdown [26]. The data were fitted with Octave package statistics.

We used the Weibull distribution, as it allows describing asymmetrical distributions better than standard deviation and can easily be used for censored data. Censored data means here that not all the lifetimes of this dataset are known, because their lifetimes exceed the length of the experiment. A dataset is left-censored when several samples failed before the measurement started, interval-censored when the measurement of the lifetimes did not occur during a time interval over the course of the test, and right-censored when the lifetimes of the samples were longer than the test procedure.

## 3. Results

### 3.1. Differences in the Time to Failure between the Samples Held at 37 °C and 57 °C

The experiment took place over the course of 458 days for the 37 °C samples and 423 days for the 57 °C samples. After that time, the experiment was stopped.

In Figure 2a, the time to failure for the samples tested at 37 °C and 57 °C is depicted. Most of the samples that were tested at 57 °C broke down between 50 and 200 days after the start of the trial, with one sample being intact after the end of the experiment and one sample taken out erroneously after 182 days. As expected, the 37 °C samples started breaking down later than those that were subjected to a test temperature of 57 °C. After 150 days, the samples started to break down, with three samples still intact after the end of the test. Both trials had in common that in between 20% and 80% of the sample failures, the failure rate was roughly constant, with roughly one sample failure every one or two weeks for 57 °C or 37 °C, respectively. We see clearly that the failures over time moved to the left for the samples that were tested at higher temperatures. In Figure 2b, we see a juxtaposition of lifetimes for the time to failure for the samples at 37 °C and a quadrupled time to failure for the samples at 57 °C. The idea was to translate the lifetimes of the samples according to the Arrhenius law from one temperature to another, because an increase of 20 °C would decrease the lifetime fourfold. While the time to failure was similar for the first about 40% of the samples and adhered to the Arrhenius law, the discrepancies arose once more than 40% of the samples failed, leading to relatively longer lifetimes at 57 °C than anticipated. This discrepancy could lead to an overestimation of the possible lifetimes based on accelerated lifetime tests.

Table 1 shows the mean and median lifetime of the samples tested at 37 °C and 57 °C. Only the median could be reliably found for both samples. As the test was right-censored, i.e., not all sample lifetimes were known, as the test was stopped before all samples failed, the mean value was subjected to uncertainties. The median of longevity for the 37 °C was as about 363 days and 138 days for the 57 °C samples. The increase in the mean lifetime was 2.175, and the increase in the median was 2.63. Both were less than the anticipated fourfold increase, expected from the Van’t Hoff rule. The standard deviation was slightly smaller for the 37 °C samples. As several samples were intact at the end of the trials, a higher standard deviation was expected at 37 °C.

The standard error equalled 25.4 and 21.2 days for the trials at 37 °C and 57 °C, respectively. The difference stemmed from the smaller number of samples at 37 °C. Considering the standard error, the mean at 37 °C could be between 380.4 and 329.6 days, and the mean at 57 °C could be between 184.2 and 141.8 days. Even considering the longest length at 37 °C and the shortest at 57 °C, only a 2.7-fold difference in lifetimes occurred, less than the expected fourfold difference.

Relative to the sample size, the number of non-failed samples was larger at the end of the trial at 37 °C. Additionally, the end of the trial was closer to the median lifetimes of the samples tested at 37 °C than at 57 °C. Thus, stronger growth can be expected for the standard deviation at 37 °C.

While the median was reliable, it is to be considered, that for the mean lifetime of the samples and the standard deviation, the end of the experiment was considered as the end of the lifetime; thus, if the experiment was conducted for a longer timespan, the mean lifetime of the samples and the standard deviation would be higher.

### 3.2. Weibull Fit

The data were analyzed and fitted via Octave and visualized via GnuPlot.

Figure 3 shows the Weibull probability plot. As the trial was right-censored, one data point was outside of the 95% confidence interval. All the other data points were well within the confidence interval. The shape factor equaled 3.673, which hinted at the aging of the samples. The scale parameter, indicating the point in time when 63% of the samples broke down, was 395.51.

Figure 4 shows the failure ratio over time and the fit for the samples that were kept at 37 °C. The red points depict the failure events, and the green line depicts the Weibull fit. As is clearly visible in Figure 4 on the left, all the points were close to the fit. As the experiment was halted ahead of the failure of all the samples after 458 days, the data were right-censored. Three samples were still intact at the end of the experiment. Due to this reason, the data points for the failure rate ended slightly above 0.8. For the middle part of the curve, the failure rate was quite constant with roughly one failure every two weeks.

A similar analysis was conducted for the data of the test at a temperature of 57 °C, as can be seen in Figure 5. On the left, the Weibull probability plot is depicted. The shape parameters equaled 1.983, and the scale parameter equaled 183.31. As one sample was still intact after 423 days at the end of the sample, the data were right-censored. Still, the last data point was within the 95% confidence interval. All the data points were within the 95% confidence interval. The coefficient of determination ρ being smaller for the 57 °C fit than for the 37 °C indicates here a slightly worse fit.

The cumulative distribution function (CDF) function was plotted in Figure 6. The discrepancy between the curve and the data points below the 80% failure rate stemmed from the relatively long lifetime of the samples that failed towards the end of the trial. Starting after 10% until about 80% of the samples failed, the failure rate was quite constant, with roughly one per week. For the most part of the curve, the failures occurred slightly before the fit predicted. In contrast to the first 80% of failed samples, the lifetimes of the longer-lasting samples were longer than predicted by the fitted curve.

Table 2 offers an overview of the scale and shape parameters of both batches. As expected the scale parameter was larger for the 37 °C samples, albeit less than four times, than for the samples at 57 °C. It was 2.16 times larger for the test at 37 °C. The shape parameter was 1.85 times larger for the 37 °C samples compared to the 57 °C samples, resulting in a much steeper rise in the sample failures at 37 °C related to the time frame. Thus, both the scale and shape parameter increased roughly twofold for the 37 °C trial compared to the 57 °C trial.

## 4. Discussion

Unlike earlier trials, where EIS was used or the resistance [9,12] was measured, here, the voltage drop over a resistance was measured via an Arduino. The chosen measurement method proved to be easy to employ and interpret and provide data that can be further used for statistical analysis. The approach allowed achieving comparably long median lifetimes of 363 and 138 days for 37 °C and 57 °C, respectively, which was longer than the other accelerated lifetime tests with 55 days [9]. The failures proved to be easily identifiable, which is not the case with the use of EIS. Additionally, the demands on the measuring devices were sufficiently low to make use of Arduinos, while allowing a continuous measurement. At the same time, it allowed for easy scalability. The provided information on the sample lifetimes could easily be used for statistics. The only problem was that the lifetimes of several samples was longer than the experiment lasted. Still, several statistical values could easily be determined, among them, the median and the scale parameter of the Weibull distribution. The mean, standard deviation, and shape parameter were subjected to incertitude, as not all the sample lifetimes were known. As this problem is widespread in similar tests, the statistics function of the used software foresaw the use of right-censored data, where the samples with longer lifetimes than the test length are treated as such. This applies to the values provided for the Weibull distribution. For the standard deviation, it can be expected that the provided values would be higher, if the test ran longer. Still, several tendencies could be identified. While the differences between the different mean values and the shape values were similar with 2.175 and 2.16, the median differed by a factor of 2.63, due to an asymmetry of the data at 57 °C. While the standard deviations at both temperatures were fairly similar, the different shape factors indicated a change in the failure events over time. Here, the shape factor provided a more clear indication of the changes than the standard deviation. The possibility of the Weibull function describing the asymmetrical distributions of lifetimes is a favorable property over the standard deviation, as it allows the Weibull function to be used on symmetrical and asymmetrical datasets and compare those. In that light, the use of the 63% value is easier, than the use of the mean value, if not all results are known. The shape parameter shows that in relation to the time frame, the change in the failure rate is more clear than with regard to standard deviation. The anticipated difference in the lifetimes was that the samples at 37 °C would be intact for four times longer than the samples at 57 °C, based on the Van’t Hoff rule, which is also the basis for the norm ASTM F 1980. As this work shows, the difference was only slightly larger than two for 20 °C difference. The use of a reaction rate constant of two would lead to anticipation of unrealistically long lifetimes. A constant of 1.47 seems more appropriate, starting from a difference of the scale parameters of 2.16. This result was surprising, considering that a reaction rate constant of two is considered as conservative according to ASTM F 1980. A long lifetime of samples is important, as for chronic applications, lifetimes of 10 years or more are desirable, and a preliminary failure should be avoided. Here, the use of the shape factor is also important, as it allows anticipating how many implants will be useful for what length of time. With larger shape factors appearing more desirable, as it allows foretelling more precisely when to expect failure. Due to the emphasis of earlier trials on the EIS method, the lack of data for similar experiments, measurements of resistance, or the use of harsher conditions, a comparison with the existing literature on polymer-based “soft” neural implants appears unrealistic.

## 5. Conclusions

Sample lifetimes of more than 400 days were achieved. The median lifetime was at 363 days for the trial at body temperature and 138 days for the trial at 57 °C. In combination with the used measuring approach, the Weibull statistics are a useful instrument that describes the lifetimes of neural implants. The Weibull distribution is better suited to describe the lifetimes of the samples than the normal distribution. The scale parameter for the test at body temperature was 396 days and 183 days at 57 °C. The change in the shape parameter, 3.673 at body temperature and 1.983 at 57 °C, indicates differences in the sample failures at elevated temperatures compared to failures at body temperature. A change of factor 2.63, 2.175, and 2.16 describes the different results of the median, mean, and shape factor, respectively. Considering, that for a temperature difference of 20 °C only a roughly 2.2-fold decrease in lifetime, according to the mean and Weibull scale factor, was measured, the use of a factor of 1.47 in equation Equation 1 seems appropriate to describe the changes in lifetime during an accelerated aging test. Thus, the Van’t Hoff rule, according to which the lifetimes decreases by a factor of two for each increase in temperature by 10 °C, would lead to overly optimistic results. That might lead to assumptions of longer lifetimes of neural implants than are realistic, with implant failure before the expected end of lifetimes. The statistics provide information on how long the samples could be used with respect to the failure probability.

In the future, this approach could be used for other test samples, such as the ones where the surfaces were treated with oxygen plasma in order to prolong their lifetimes. Moreover, future analyses could make use of the Arrhenius equation in order to estimate the activation energy for sample failures of various samples. Here, the influence of material treatment, i.e., the use of oxygen plasma, should be considered too. The applicability of the Weibull statistics to other designs, like wire electrodes, or thin-film-based electrodes with oxide layers, could be examined. And whether the application of the Weibull statistics on in vivo results is possible should be examined. Also, the transferability of the measured results from test samples to neural samples should be conducted with caution, as size effects may apply.

## Figures and Tables

**Figure 1 sensors-23-06263-f001:**
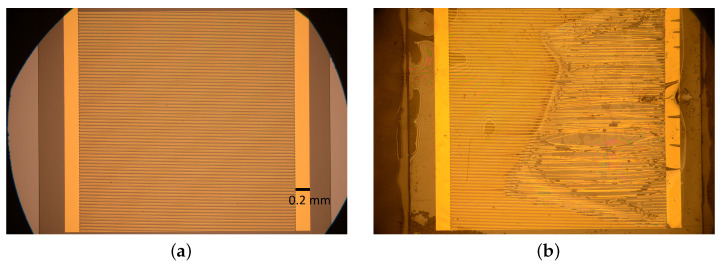
Samples before and after the long-term test. (**a**) The state of a sample after production and before the trial. (**b**) The state of a sample after failure.

**Figure 2 sensors-23-06263-f002:**
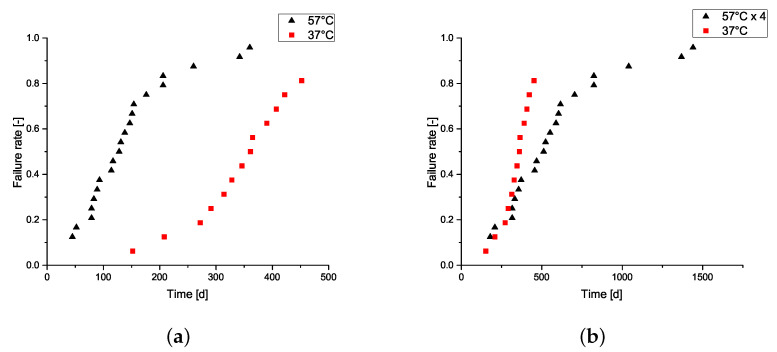
Time to failure of the samples. (**a**) Failure ratio over time for both temperatures. (**b**) Failure ratios over time, with the failure times at 57 °C multiplied by four to show the difference between the times to failure.

**Figure 3 sensors-23-06263-f003:**
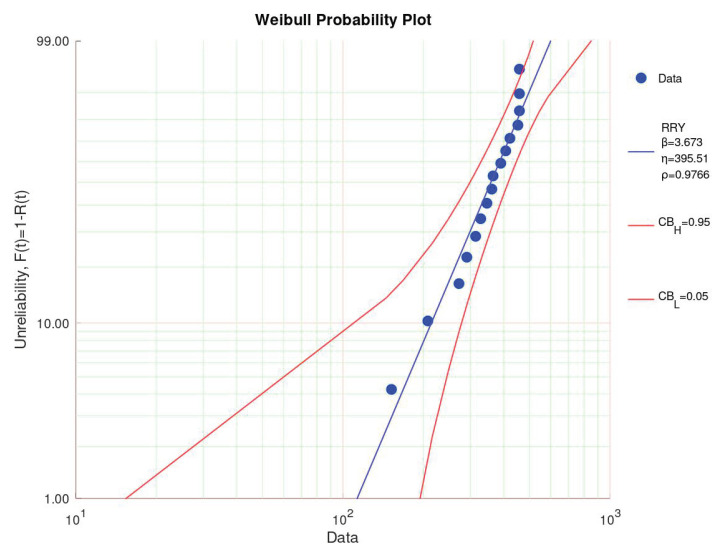
Weibull probability plot at 37 °C.

**Figure 4 sensors-23-06263-f004:**
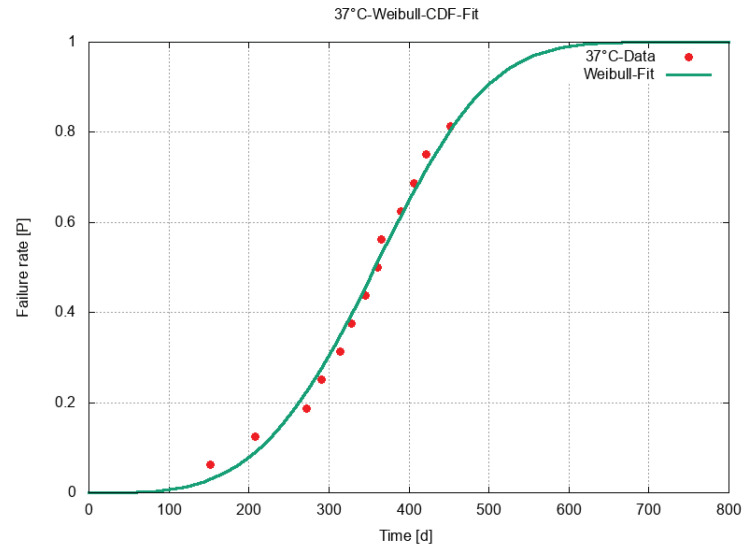
Data points and the Weibull-CDF fit at 37 °C. The end of the test was after 458 days. Three samples were still intact at that time.

**Figure 5 sensors-23-06263-f005:**
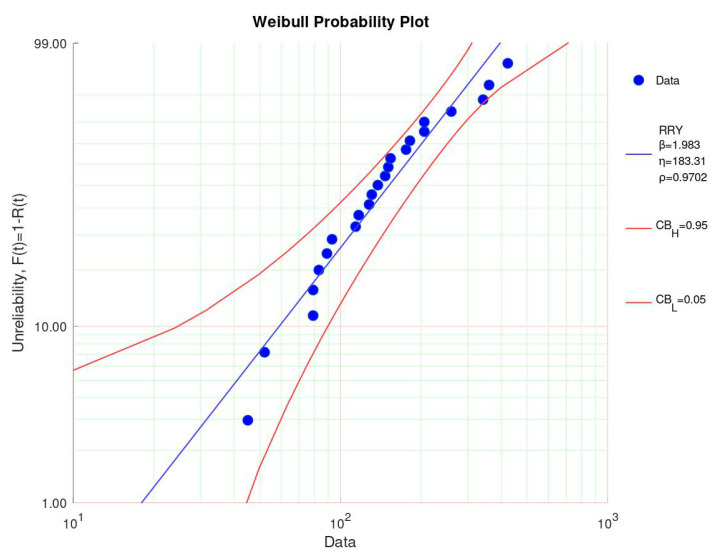
Weibull probability plot at 57 °C.

**Figure 6 sensors-23-06263-f006:**
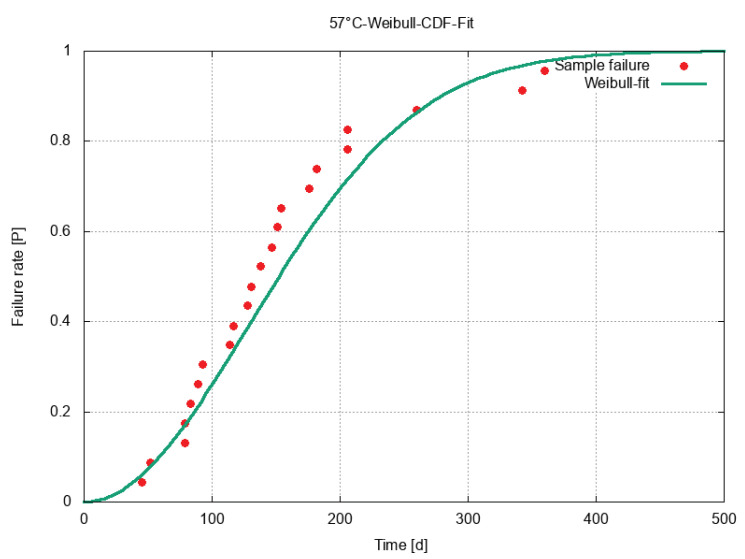
Data points and the CDF fit at 57 °C. The test stopped after 423 days, with one sample still intact.

**Table 1 sensors-23-06263-t001:** The mean and median times for different temperatures. The mean, standard deviation, and standard error, marked with a *, are subject to strong uncertainty.

Value	37 °C	57 °C	Difference
Median	363	138	2.63
Mean *	355	163	2.175
Std. Dev. *	91.7	99.3	0.923
Std. Err. *	25.4	21.2	-

**Table 2 sensors-23-06263-t002:** Shape and scale parameters for both temperatures.

Parameter	37 °C	57 °C	Difference
Scale parameter	395.51	183.31	2.16
Shape parameter	3.673	1.983	1.85

## Data Availability

Not applicable.

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
