# Peer review of "Analysis of the Lifetime of Neural Implants Using In Vitro Test Structures"

_sensors, 2023, doi:10.3390/s23146263_

Round 1
Reviewer 1 Report
Dear Authors,
I studied your manuscript entitled "Lifetime of neural implants analyzed using in vitro test structures". Some spaces need to be improved in terms of journal quality. I recommend a major revision before further consideration for publication in the Sensors.
1) In the Introduction section, please provide more information on the significance and potential impact of the study. Compared to previous work by the authors, what new insights does this study provide into the aging of neural implants?
2) Could you explain how the results of this study could contribute to safer and more reliable neural implants for humans, as well as the potential clinical applications of accelerated lifetime tests?
3) The Van't Hoff rule may have limitations when calculating the possible lifetime of neural implants, and how might alternative approaches be used to validate the results?
4) According to Table 2, how do the scale and shape parameters for the samples tested at 37°C compare to those tested at 57°C?
Author Response
Dear Reviewer,
Thank you for your reply.
Kind regards
Jürgen Guljakow

Reviewer 2 Report
The manuscript entitled, ‘Lifetime of neural implants analyzed using in vitro test structures’ discussed Lifetime of neural implants. The article should be modified according to the following points;
1. The introduction is not clearly discussed why this work is worth. Better to emphasize the novelty.
2. It will be better if the figure 1 has the scale bar.
3. Author wrote “test is right-censored”; what does it significant here?
4. Why 20degree temperature gap was monitored? Any special reason? Make arguments on it.
Author Response
Dear Reviewer,
Thank you for your reply,
Regards
Jürgen Guljakow

Reviewer 3 Report
The reviewed manuscript focused on the lifetime of samples designed to mimic the behavior of neural implants. These samples consisted of interdigitated gold strands encapsulated in polyimide and were subjected to soaking in ringer solution. The authors created two batches for the study, with one batch soaked at 37°C and the other at 57°C. The application of voltage and subsequent measurement aimed to identify instances of failure throughout the duration of the experiment. I am sure the study deserves readers' attention and can be published. However, some major concerns need to be addressed before accepting the paper for publication to improve the readability and clarity of the manuscript:
1- Please consider reviewing the abstract and highlighting the novelty. The abstract should contain answers to some questions, what problem was studied and why is it important? and what conclusions can be drawn from the results?
please provide specific results and not generic ones. Please use numbers or % terms to clearly shows the results of your experimental work.
2- The introduction is poor and needs a lot of enhancement. The introduction is very short and brief. Although we are now in the middle of 2023, the authors just mentioned one reference in 2023 (for the same authors) and two references in 2022. I think it needs to be updated.
3- The authors mentioned at the end of the introduction what they are focusing in the paper in brief. I think the paragraph at the end of the introduction should clarify the importance of the work, the methodology of the work, and notes about expected outcomes not specific results.
4- In the experimental section, the authors should mention the companies of the utilized materials with a relation to their countries.
5- The sample preparation lacks of graphical images or schematics that show the sample preparation technique. This is the section on experimental work and the authors should provide sufficient graphical information for the readers to better understand their work and what was done in it.
6- The authors mentioned the mean value without referring to the standard error.
7- Did the authors designed their tube dimensions according to a standard used with a quasi-static uniform axial compression test? If yes, please refer to that with a reference.
8- When discussing and comparing results, quantitative information should be given instead of only qualitative information.
9- Some of the results are merely described and is limited to comparing the experimental observation and describing results. The authors are encouraged to include a more detailed results and discussion section and critically discuss the observations from this investigation with existing literature.
10- Conclusions need to highlight the outcomings of the scientific paper. Furthermore, the authors didn't mention their future work at the end of the conclusions.
Please, read the text carefully before the next submission of the paper.
Need minor revision
Author Response
Dear Reviewer,
Thank your for your reply,
Regards
Jürgen Guljakow

Round 2
Reviewer 1 Report
Dear Authors,
Thank you for considering my comments and for the revision which enhances both the clarity and the relevance of your work. I have recommended the publication of your article as is.
Author Response
Dear Reviewer,
Thank you for your remarks,
Regards
Jürgen Guljakow
Reviewer 2 Report
This can be accepted in its present form
Author Response
Dear Reviewer,
Thank you for your remarks.
Regards
Jürgen Guljakow
Reviewer 3 Report
I appreciate the authors for incorporating my recommendations and improving their paper. While the paper has indeed shown significant enhancements, I would like to suggest a few minor adjustments that could be considered for further improvement.
It is important to update the introduction section to reflect the most current and relevant information. Additionally, addressing the issue of mean results will enhance the certainty and reliability of the paper's outcomes. Lastly, in the conclusion section, it would be appropriate to suggest future work for other researchers rather than indicating that the same authors should conduct it. These adjustments will further strengthen the paper and improve its overall quality.
better than before
Author Response
Dear Reviewer,
Thank you for your remarks.
regards
Jürgen Guljakow
